# Serum Follicle-Stimulating Hormone Levels are Associated with Cardiometabolic Risk Factors in Post-Menopausal Korean Women

**DOI:** 10.3390/jcm9041161

**Published:** 2020-04-18

**Authors:** Eun-Soo Jung, Eun-Kyung Choi, Byung-Hyun Park, Soo-Wan Chae

**Affiliations:** 1Clinical Trial Center for Functional Foods, Chonbuk National University Hospital, Jeonju, Jeonbuk 54907, Korea; esjung@jbctc.org (E.-S.J.); ekchoi1477@gmail.com (E.-K.C.); 2Department of Biochemistry and Molecular Biology, Chonbuk National University Medical School, Jeonju, Jeonbuk 54896, Korea

**Keywords:** FSH, menopause, cardiometabolic risk, metabolic syndrome

## Abstract

Menopause compounds many cardiometabolic risk factors through endogenous estrogen withdrawal. This study aimed to find the association between serum follicle-stimulating hormone (FSH) levels and cardiometabolic risk factors in post-menopausal Korean women. A total of 608 post-menopausal women from eight randomized double-blind, placebo-controlled clinical trials on menopause during the year 2012–2019 were analyzed. Cardiometabolic risk factors such as body mass index, waist circumference, systolic blood pressure, fasting glucose, triglycerides (TG), high density lipoprotein-cholesterol (HDL-C), and TG/HDL-C ratio were significantly improved as the FSH quartiles increased. Metabolic syndrome (MetS) and the number of components of MetS decreased as FSH quartiles increased. In regression analysis, FSH level was negatively associated with cardiometabolic risk factors including body mass index, body weight, waist circumference, fasting glucose and TG, while it was positively associated with HDL-C. The odds ratio of MetS in the first quartile of FSH was 2.682 compared with that in the fourth quartile of FSH in a logistic regression model. Serum FSH levels had a negative correlation with cardiometabolic risk factors in post-menopausal Korean women, suggesting that a low FSH can be a predictor for cardiovascular disease in post-menopausal women.

## 1. Introduction

Cardiometabolic risk is a condition in which the possibilities of development of atherosclerotic cardiovascular disease (CVD) and diabetes mellitus (DM) increase. This concept covers traditional risk factors such as hypertension, dyslipidemia, and smoking, as well as emerging risk factors, such as abdominal obesity, inflammatory profiles, and ethnicity [1,2].

Women’s cardiometabolic risk significantly increases after shifting into menopause [3], and CVD is generally the leading cause of mortality in post-menopausal women [4]. A recent meta-analysis showed that post-menopausal women had about 1.5~4 times higher cardiometabolic risk and 3.5 times higher risk of development of metabolic syndrome (MetS) than premenopausal women [5]. Cardiometabolic risk is particularly prevalent in patients diagnosed as having MetS, and it is not only associated with aging but also, at least in part, related to decreased levels of ovarian hormones during the menopausal transition and beyond [3]. Also, the higher odds of development of MetS are known due to the larger reduction in estrogen, than androgens, along with increased LDL-cholesterol (LDL-C) levels and reduced HDL-cholesterol (HDL-C) levels during the post-menopausal period [6,7,8]. 

However, in general, associations between endogenous estrogens and CVD risk were not consistent. Some studies have confirmed the expected cardio-protective effects of estrogens, while others failed to do that [9,10], and there were even studies reporting that endogenous estrogens may be associated with higher CVD risk [11,12] and type 2 diabetes [13,14]. Furthermore, recent studies have gradually reported associations between follicle-stimulating hormone (FSH) and cardiometabolic risk factors in post-menopausal women. These studies have shown that lower FSH is associated with prediabetes and diabetes [15] and higher prevalence of MetS [16,17]. Therefore, a low level of FSH is even suggested as a risk factor or a biomarker for CVD in post-menopausal women [18].

However, the role of FSH in the development and progression of CVD is still unknown, and in particular, the association between FSH and cardiometabolic risks has not been studied in post-menopausal Korean women. In this study, we addressed the hypothesis that serum FSH level is associated with cardiometabolic risk factors in post-menopausal Korean women.

## 2. Subjects and Methods

### 2.1. Data Collection

Previous studies, which were conducted at the Clinical Trial Center for Functional Foods (CTCF2) of Chonbuk National University Hospital, were eligible for re-analysis if they contained data on endogenous sex hormones and cardiometabolic risk factors using prospectively collected blood samples from post-menopausal women. A total of eight eligible studies were identified, which were randomized double-blind, placebo-controlled clinical trials (RCTs) for evaluating effects and safeties of functional food materials on improvements of menopausal symptoms in post-menopausal Korean women, conducted during the years 2012–2019.

All of the RCTs were conducted according to the Helsinki Declaration and the Guideline for Good Clinical Practice by the International Conference on Harmonization (ICH GCP). The study protocols for both the RCTs and this re-analysis were approved by the Institutional Review Board (IRB) of Chonbuk National University (IRB No. JBNU 2019-12-004).

Eligible participants of this study were healthy post-menopausal women with elevated FSH and low estradiol (E2) levels, who had complained of menopausal symptoms and had the last menstrual period (LMP) at least 12 months prior to the enrollment. Participants who showed levels more than 40.0 mIU/mL of FSH and estradiol less than 30 pg/mL of estradiol were defined to be of sufficiently low ovarian hormonal status for including to this analysis (according to the 2011 Stages of Reproductive Aging Workshop +10 recommendation [19]).

Subjects who had the following conditions were excluded from the RCT studies: (i) taking any hormonal replacement therapy, (ii) having severe menopausal symptoms which needed to be treated, (iii) having any other current illnesses or any history of chronic medical diseases within the past 3 years, (iv) having participated in any other clinical trials with investigative products within the past 2 months, (v) being allergic or hypersensitive to any of the ingredients in the investigative products, and (vi) having conditions which could interfere with successful participation in the studies or which could risk a subject’s safety in the opinion of the investigators: women during pregnancy or breast feeding; history of alcohol or drug abuse; abnormal laboratory tests; medical or psychological conditions.

The participants were recruited around the city region through local advertisements during the years 2012–2019. They pre-checked the volunteers’ eligibility by telephone interviews using a structured questionnaire and then were scheduled to visit the hospital for a screening test. All volunteers gave written informed consent before participating in the studies.

### 2.2. Biochemical and Hormone Analyses

Blood samples were collected after an overnight fasting and centrifuged on the spot in 1 h after collection. Serum lipid profiles including total cholesterol, TG, HDL-C and LDL-C were measured by a Hitachi 7600-110 analyzer (Hitachi High-Technologies Corporation, Tokyo, Japan). Serum E2 and FSH were measured by Modular Analytics (E170) (Roche, Basel, Switzerland). Serum TSH was measured by ADVIA Centaur XP Immunoassay System (Siemens Healthcare GmbH, Erlangen, Germany). Hematology (CBC) and other laboratory tests (total protein, albumin, ALP, ALT, AST, γ-GTP, total bilirubin, BUN, and creatinine, uric acid, total calcium, calcium, glucose, urine specific gravity, urine pH,) were measured using the Sysmex XE-5000^TM^ (Sysmex Corporation, Kobe, Japan) and ADVIA^®^2400 chemistry system (SIEMENS, Munich, Germany), respectively.

### 2.3. Lifestyle Questionnaires

Physical activity was evaluated by using the Global Physical Activity Questionnaire (GPAQ). The GPAQ comprises 16 items that quantify the physical activity levels of a normal active week for each participant. Metabolic equivalent tasks (METs ) were used to express the intensity of physical activity, and classified into three intensity levels: vigorous (8 METs), moderate (4 METs) and inactivity (1 MET) [20].

For dietary assessment, dietary intake data was collected from food records of each participant. The food intake and contributions of energy and nutrient intakes were calculated by using the CAN-pro 4.0 (Korean Nutrition Society, Seoul, Korea). 

### 2.4. Definition of Variables

To define cardiometabolic risk factors, we used definitions as proposed by the National Cholesterol Education Program (NCEP) Adult Treatment Panel III (ATPIII) [2,21], with minor modification as stated in the Western Pacific Regional Office of the World Health Organization (WHO) [22] and an Asia-Pacific perspective [23]: for overweight and obesity, body mass index (BMI) measures of 23 to 29.9 kg/m^2^ and ≥30 kg/m^2^, respectively; for high-fasting glucose, fasting glucose level ≥ 100 mg/dL; for hypertension, systolic blood pressure ≥ 130 mm Hg and diastolic blood pressure ≥ 85 mm Hg; for dyslipidemia, triglycerides (TG) ≥ 150 mg/dL or HDL-C < 50 mg/dL. The diagnosis of MetS are as follows: three or more than of waist circumference (WC) ≥ 80 cm, fasting glucose ≥ 100 mg/dL, TG ≥ 150 mg/dL, HDL-C < 50 mg/dL, and systolic pressure ≥ 130 or diastolic pressure ≥ 85 mmHg.

### 2.5. Statistical Analysis

Statistical analysis was performed using SAS Statistics, version 9.4 (SAS Institute, Cary, NC, USA). Continuous variables are presented as the mean ± standard deviation (SD) and categorical variables are presented as a percentage (%). Participants were divided into quartiles according to the serum FSH levels, with an equal number of participants in each group. A Chi-square (χ^2^) test was then performed to determine differences between the frequencies of categorized variables between the FSH quartiles groups. Analysis of variance (ANOVA) was used for assessing differences of continuous measures between the FSH quartiles groups. *p* for trend was calculated by ANOVA (Jonckheere–Terpstra) and χ^2^ (Cochran–Armitage) tests.

Association between FSH and cardiometabolic risk factors was analyzed using linear regression model with each measure as the dependent variable and FSH as the independent variable. The regression model was adjusted for age, body weight, and estradiol. The association between FSH and MetS (categorical variables) was assessed by a logistic regression. Results were expressed as odds ratios with 95% confidence interval (CI). All analyses were 2-sided; *p*-values of less than 0.05 were considered statistically significant.

## 3. Results

### 3.1. General Characteristics of the Study Participants

In all, 609 post-menopausal women among total 692 volunteers who had participated in the eight RCTs were included in this analysis (Figure 1). General anthropometric and laboratory characteristics of the study participants are shown in Appendix A. The mean age of all 609 subjects was 53.51 ± 2.94 years old and the mean Kupperman Index (KI) score was 34.21 ± 7.17 point. Among participants, the number of hypertensive patients with antihypertensive medications was 21 (3.45%). The mean serum FSH level was 79.67 ± 24.36 mIU/mL (reference range: 25.8–134.8 mIU/mL) and estradiol level was 7.99 ± 9.38 pg/mL (reference range: 5.0–138.0 pg/mL). Overall, the study participants were characterized by post-menopausal women having moderate menopausal symptoms with overweight (23.54 ± 2.69 kg/m^2^ of BMI), central obesity (0.88 ± 0.06 of waist to hip ratio), and dyslipidemia (203.78 ± 32.40 mg/dL of total cholesterol levels).

Anthropometric and laboratory characteristics were categorized according to FSH quartiles (Table 1). Each of the ranges of FSH quartiles were as follows: ≤62.20 mIU/mL for the first quartile (Q1), 62.21~76.87 mIU/mL for the second quartile (Q2), 76.88~94.00 mIU/mL for the third quartile (Q3), and ≥94.01 mIU/mL for the fourth quartile (Q4). A trend analysis showed a declining trend in age, KI score and body weight as FSH quartiles increased (*p* for trend <0.05). In addition, the levels of age and body weight of the Q1 were significantly higher than those of the Q4, while there were no significant differences in alcohol intake, height, physical activities, etc. between the FSH quartiles groups. The results of an ANOVA analysis of the anthropometric and laboratory characteristics were not attenuated after adjustment for age, body weight and estradiol all together.

All values of the laboratory variables of the four FSH quartiles were within the limits of normal reference ranges though the values of RBC and hemoglobin were statistically different between the Q1 and the Q2, which were meaningless in terms of clinical standpoint (Table 2). 

### 3.2. Diet and Physical Activity of the Study Participants 

Amounts of energy and nutrient intakes of the four FSH quartiles are summarized in Appendix A. There were no significant differences in energy and nutrients intakes among the four FSH quartile groups. Also, physical activity of the four FSH quartiles were not different among the four FSH quartile groups.

### 3.3. Cardiometabolic Risk Factors of the Study Participants

BMI, WC, systolic blood pressure, fasting glucose, TG, and TG/HDL-C ratio showed a significant decline, while total cholesterol, HDL-C, and LDL-C showed a significant increment as the FSH quartiles increased on a trend analysis of the cardiometabolic risk factors categorized according to FSH quartiles (*p* for trend < 0.05) (Table 1).

As well, BMI, WC, systolic blood pressure, fasting glucose, TG, and TG/HDL-C ratio in the Q1 were higher than those of the Q4 (*p* < 0.05) on an ANOVA analysis, whereas total cholesterol, HDL-C, and LDL-C in the Q1 were lower than those of the Q4 (*p* < 0.05). In addition, *p* values of dependent variables except those of WC, systolic blood pressure showed similar results after adjustment for age, body weight and estradiol. 

### 3.4. Association between Follicle-Stimulating Hormone (FSH) Levels and Cardiometabolic Risk Factors

FSH level was inversely related with cardiometabolic risk factors as follows after adjustment for age, body weight, and estradiol: BMI (B −0.024; 95% CI, −0.033 to −0.015, *p* < 0.01), body weight (B −0.067; 95% CI, −0.091 to −0.043, *p* < 0.01), WC (B −0.023; 95% CI, −0.044 to −0.001, *p* < 0.05), and fasting glucose (B −0.028; 95% CI, −0.054 to −0.003, *p* < 0.05), while the FSH level was positively correlated with HDL-C (B 0.063; 95% CI, 0.014 to 0.128, *p* < 0.05) and total cholesterol (B 0.142; 95% CI, 0.033 to 0.252, *p* < 0.05) (Figure 2A).

Elevation of serum thyroid-stimulating hormone (TSH) is common in post-menopausal women [24], and serum levels of TSH associate positively with cardiometabolic risk factors even in post-menopausal women with normal thyroid function [25,26]. Therefore, we analyzed the association between FSH and cardiometabolic risk factors after adjustment for age, body weight, and TSH. Results show that FSH was also inversely related with cardiometabolic risk factors: BMI (B −0.028; 95% CI, −0.038 to −0.017, *p* < 0.001), body weight (B −0.077; 95% CI, −0.105 to −0.048, *p* < 0.001), WC (B −0.035; 95% CI, −0.059 to −0.011, *p* < 0.01), and fasting glucose (B −0.031; 95% CI, −0.062 to 0.001, *p* = 0.055), while the FSH level was positively correlated with total cholesterol (B 0.178; 95% CI, 0.042 to 0.313, *p* < 0.05) and LDL-C (B 0.164; 95% CI, 0.013 to 0.315, *p* < 0.05) (Figure 2B). Overall, the results indicated that higher FSH level was associated with lower cardiometabolic risk factors.

### 3.5. Association between Estradiol Levels and Cardiometabolic Risk Factors

Because estrogen is the primary hormone responsible for CVD in menopausal women [9], we further analyzed the association between estrogen levels and cardiometabolic risk factors. Estradiol level was inversely related to WC (B −0.08; 95% CI, −0.038 to −0.119, *p* < 0.05) after adjustment for age, body weight, and FSH (Figure 3A). When the analyses were repeated after adjustment for age, body weight, and TSH, estradiol level was positively correlated with body weight (B 0.255; 95% CI, 0.111 to 0.399, *p* < 0.001) and BMI (B 0.096; 95% CI, 0.044 to 0.148, *p* < 0.001) and inversely related to HDL-C (B −0.437; 95% CI, −0.793 to −0.080, *p* < 0.05) (Figure 3B).

### 3.6. Association between FSH Levels and Metabolic Syndrome (MetS)

The prevalence of MetS and the number of its components also showed a declining trend as the FSH quartiles increased (*p* for trend <0.05) and had a significant difference among groups on ANOVA analysis (*p* < 0.05, Table 3). The odds ratios of MetS and its components, WC, of the Q1 relative to those of the Q4 were 2.391 (95% CI, 1.047 to 5.459, *p* < 0.05), 2.606 (95% CI, 1.136 to 5.975, *p* < 0.05), respectively. Odds of variables were maintained after adjustment with age, obesity, and estradiol, although those were slightly attenuated (Table 4 and Table 5, Appendix A).

## 4. Discussion

The main goal of the present study was to find the association between serum FSH levels and both cardiometabolic risk factors and MetS in post-menopausal women. The study also aimed to find the odds of cardiometabolic risk factors and MetS depending on their serum FSH status. We found that FSH levels of the post-menopausal women were significantly associated with cardiometabolic risk factors and the prevalence of MetS or its components after adjustment for age, estradiol, and body weight. Lower FSH levels resulted in increasing odds of cardiometabolic risk and MetS.

In order to explain the notable increase in CVD risk after menopause, while it has been hypothesized traditionally that women’s cardiovascular health are protected by their endogenous estrogens prior to menopause [27,28,29,30], the role of estrogen after menopause and during the menopausal transition is far less apparent, and the marked heterogeneity among the studies even made uncertain the question as to whether menopause affects the incidence of CVD has been answered [3,9,10,11,12,13,14]. Furthermore, the Study of Women’s Health across the Nation (SWAN) study reported that E2 trajectories over the menopausal transition were not uniform across the population of women: The women with a normal weight tend to have a slow decline in E2 trajectories, and race/ethnicity and body mass index affect the trajectory of E2 change over the menopausal transition [31]. Therefore, other hypotheses apart from the estrogen theory would be required to explain the higher CVD risk after menopause.

In this study, FSH and estrogen levels of the post-menopausal women had a significant relation to cardiometabolic risk factors and the prevalence of MetS or its components (Table 4 and Table 5, Figure 2). There were other studies which have reported the associations between serum FSH and cardiometabolic risk in post-menopausal women: low level of FSH in post-menopausal women has been suggested as a risk factor or a biomarker for CVD [18] and speculated as a predictive biomarker of glucose metabolism from the fact of its associations with prediabetes and diabetes [15]. Also, FSH was suggested as a biomarker to assess the probability of MetS better than C-reactive protein, leptin [16] and sex hormone binding globulin [17] in post-menopausal women. In contrast, there were several studies which have demonstrated deleterious effects of FSH on CVD risk measures [32,33].

To explain the beneficial effect of FSH on cardiometabolic risk, a causal relationship between serum FSH and cardiometabolic risk has been suggested through its relation to obesity: Serum FSH level is lower in obese women [34,35] and weight loss elevates FSH levels in overweight post-menopausal women [36]. Although these studies suggest obesity as a potential mediator of the relationship between FSH and cardiometabolic disease, findings about the effects of FSH on cardiometabolic risk in post-menopausal women are inconsistent, and the pathophysiology that underlie this association is not well determined yet. Therefore, it is necessary to further investigate mechanisms by which FSH affect cardiometabolic risk.

In our study, age and body weight tended to decrease significantly as the FSH quartiles increased (*p* for trend = 0.0014), and serum estradiol level of the Q1 was significantly higher compared to those of the Q2 and Q3 (*p* = 0.0117) (Table 1). Thus, we questioned whether the heterogeneity of the age, body weight and estrogen levels among the four FSH quartiles, as confounders, might affect the results. After an adjustment for both age and estrogen in the linear regression analysis, the associations between FSH and cardiometabolic risk factors remained significantly the same as before the adjustment, except for a variable of systolic blood pressure. Also, the associations remained significant, except for two variables including systolic blood pressure and TG among the cardiometabolic risk factors, even after adjustment for not only age and estrogen but also body weight (Figure 2). Odds of the prevalent MetS were also maintained significantly after adjustment for age, estradiol, and body weight (Table 4). The findings suggest that the association between FSH and cardiometabolic risk factors may be independent of age, estrogen, and body weight, and lower FSH levels independently increase odds of the development of cardiometabolic risk and MetS in menopausal women.

In line with these results, recent studies have reported FSH and/or FSH receptor activity in extra-gonadal tissues including bone, placenta, endometrium, liver, and blood vessels, suggesting other functions beyond its reproductive ones [37,38]. In addition, FSH is known to increase neovascularization in the umbilical vein [39] and inhibits hepatic cholesterol biosynthesis [40] via the direct effect, and indirectly in granulosa cells [41] and ovarian cancer cells [42] by inducing vascular endothelial growth factor (VEGF) production. Moreover, plasma VEGF levels are predictive of lower 10-year CVD risk [43]. Conversely, there are results opposed to the protective effects of FSH on cardiometabolic risk [32,33]. A study reported that FSH interacted with FSH receptors in hepatocytes and reduced LDL receptor levels, which subsequently resulted in increasing blood LDL-C in mice [44]. Similarly, other studies have found that FSH promoted lipid biosynthesis in adipose tissue and visceral fat accumulation by upregulating FSH receptor mRNA expression and the G*ai*/Ca^2+^ /cAMP-response-element-binding protein pathway [45,46]. These results seem to be contradictory to hypotheses about the benefit of FSH on cardiometabolic risk factors and in need of further investigations to confirm the hypothesis.

The study had some strengths. First, it is the first study to investigate the association between FSH level and cardiometabolic risk in post-menopausal Korean women. Second, the data came from the RCTs which had followed very strict protocols with the same format. All eight RCTs had aimed at the same object, which were intended to verify the statistical superiority on improvement of menopausal symptoms in post-menopausal women. Importantly, serum levels of FSH and estradiol were also added to the inclusion criteria for judging menopausal status of the subjects even though self-report of the subjects on their menopausal symptoms is enough to classify their own menopausal status [47]. In addition, the subjects who had their LMP more than four years ago prior to a screening test were not enrolled in RCTs because average duration of post-menopausal symptoms, especially vasomotor symptoms which are the most common symptoms making mid-life women visit clinics, are four years [48]. Researchers had regarded that hot flushes might be due to conditions other than menopause if post-menopausal symptoms of the subjects had persisted longer time even though there are 10% of the women reporting hot flushes 12 years after the LMP [44]. Moreover, anthropometric measurements and questionnaires were completed by the same trained research group with strong quality control standards. Third, we assessed the lifestyle of participants such as food intake and physical activity just prior to both the start and end of the RCTs in order to make sure that the participants had retained their usual lifestyle unchanged. We found that there were no differences in food intake and physical activity through the intervention period among the four FSH quartiles, which indicated that the lifestyle of the participants did not affect the results of the study.

However, our study also has some limitations because it is not a prospective study. Therefore, first, we could not draw causal relationships between FSH and cardiometabolic risk factors. Second, we only measured FSH and E2 for a single time. However, this may not significantly affect the results because FSH and E2 are considered to be stable about 2 years after LMP [35].

Taken together, serum FSH levels, independent of age, estradiol, and body weight were inversely associated with cardiometabolic risk factors in post-menopausal women. Also lower FSH levels independently increased odds of the prevalent cardiometabolic risk and MetS. These results indicated that a low FSH might be an independent predictor for CVD in post-menopausal women and it is worth exploring further whether FSH is a protective biomarker of CVD risk in post-menopausal women and the mechanism which mediates the association between FSH and CVD risk.

## Figures and Tables

**Figure 1 jcm-09-01161-f001:**
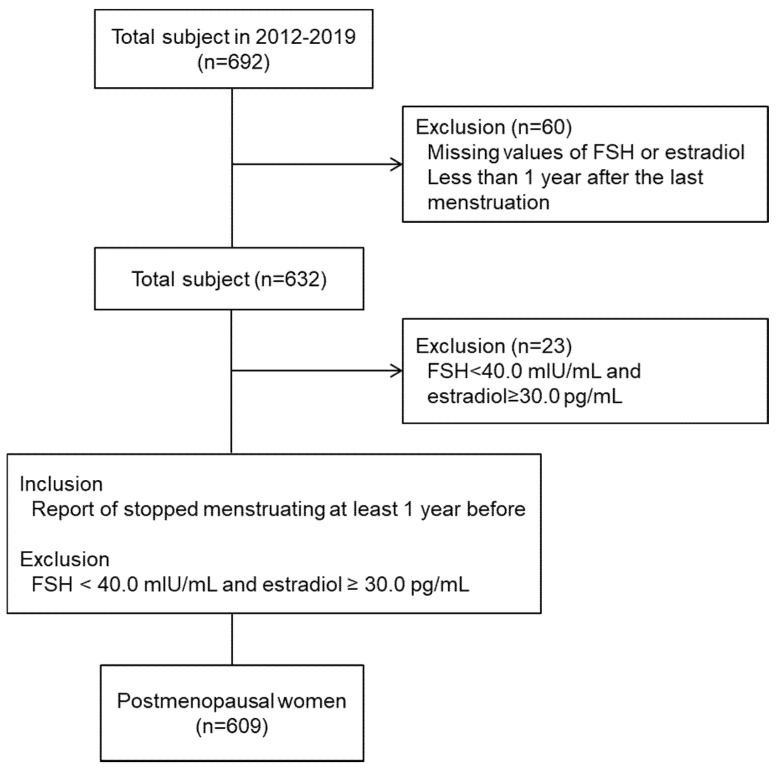
Flow diagram of study selection process.

**Figure 2 jcm-09-01161-f002:**
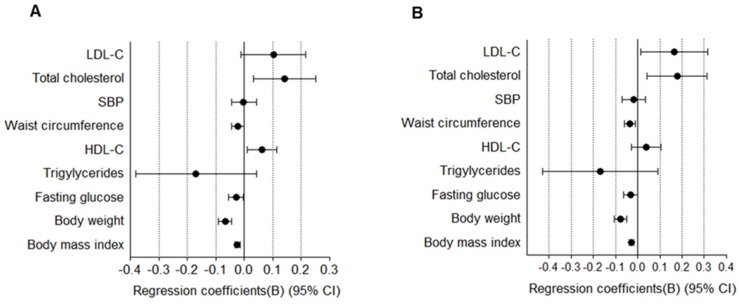
Associations between follicle-stimulating hormone (FSH) and cardiometabolic risk factors in post-menopausal women. (**A**) The model was adjusted for age, estradiol, and body weight. (**B**) The model was adjusted for age, thyroid-stimulating hormone (TSH), and body weight. Data were analyzed using multivariate regression models with each measure as the outcome and follicle-stimulating hormone as the explanatory variable.

**Figure 3 jcm-09-01161-f003:**
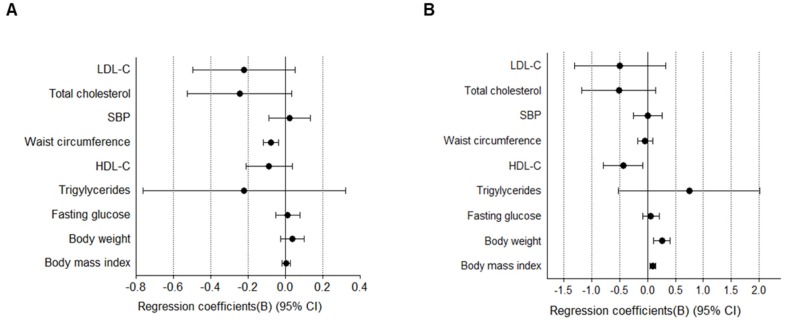
Associations between estradiol and cardiometabolic risk factors in post-menopausal women. (**A**) The model was adjusted for age, FSH, and body weight. (**B**) The model was adjusted for age, TSH, and body weight. Data were analyzed using multivariate regression models with each measure as the outcome and estradiol as the explanatory variable. FSH, follicle-stimulating hormone; TSH, thyroid-stimulating hormone.

**Table 1 jcm-09-01161-t001:** General characteristics and cardiometabolic risk factors of study participants by quartiles of follicle-stimulating hormone (FSH).

Variables		Follicle-Stimulating Hormone (mIU/mL)				*p* for Trend ^1^	*p* ^2^	*Adj. p* ^3^
		Q1 (≤62.20)	Q2 (62.21~76.87)	Q3 (76.88~94.00)	Q4 (≥94.01)			
Number		153	152	152	152	-	-	-
Age (years)		53.86 ± 2.99^a^	53.80 ± 3.10^ab^	53.41 ± 2.84^ab^	52.96 ± 2.77^b^	0.0014	0.0277	0.0104
Menopausal age (years)		50.81 ± 2.99	50.00 ± 3.40	50.23 ± 2.99	49.93 ± 3.32	0.2500	0.4272	0.6150
Alcohol	(N, %)	48(37.50%)	57(40.14%)	52(37.68%)	55(37.73%)	0.9574	0.9679	-
	(g/week)	16.01 ± 14.22	20.98 ± 25.58	18.23 ± 22.73	20.08 ± 24.89	0.6350	0.6904	0.7788
Blood pressure–lowering drugs (N, %)		5(3.27%)	3(1.97%)	7(4.61%)	6(3.95%)	0.4808	0.6297	-
KI		35.73 ± 6.68^a^	33.77 ± 7.52^ab^	33.98 ± 7.16^ab^	33.38 ± 7.13^b^	0.0157	0.0436	0.0503
MRS		33.16 ± 7.80	31.54 ± 8.29	32.20 ± 7.60	30.66 ± 8.19	0.1252	0.2088	0.3533
Estradiol (pg/mL)		10.29 ± 15.64^a^	7.02 ± 6.66^b^	6.63 ± 4.44^b^	7.99 ± 5.98^ab^	0.9445	0.0117 ^†^	0.0010
Height (cm)		158.01 ± 5.00	157.53 ± 4.71	156.96 ± 4.86	156.97 ± 5.58	0.0863	0.2066	0.9461
Weight (kg)		61.01 ± 7.12^a^	58.50 ± 7.96^b^	56.93 ± 7.22^b^	56.83 ± 6.28^b^	<.0001	<.0001	<.0001
Physical activity (METs/week)		2438.75 ± 2581.00	2184.47 ± 2271.13	2768.22 ± 2915.91	2778.69 ± 3972.33	0.7957	0.3987	0.4659
BMI (kg/m^2^)		24.45 ± 2.73^a^	23.56 ± 2.90^b^	23.08 ± 2.46^b^	23.08 ± 2.39^b^	<.0001	<.0001	<.0001
WC (cm)		85.63 ± 7.79^a^	83.26 ± 8.40^ab^	82.48 ± 7.16^ab^	81.50 ± 6.76^b^	0.0010	0.0067	0.2779
HC (cm)		94.24 ± 5.61	92.90 ± 3.59	92.37 ± 4.23	92.53 ± 3.71	0.1547	0.3051	0.8876
WHR		0.89 ± 0.05	0.88 ± 0.06	0.87 ± 0.06	0.88 ± 0.05	0.3693	0.6218	0.7451
SBP (mmHg)		122.17 ± 13.02^a^	118.49 ± 13.11^ab^	117.58 ± 12.47^b^	119.47 ± 14.25^ab^	0.0488	0.0167	0.2051
DBP (mmHg)		77.79 ± 10.46	77.67 ± 9.91	76.22 ± 10.31	77.68 ± 10.17	0.4627	0.4867	0.4921
Pulse (BPM)		70.61 ± 8.66	70.58 ± 8.66	71.30 ± 8.58	71.41 ± 7.83	0.2007	0.7400	0.9339
Glucose (mg/dL)		89.24 ± 8.06^a^	88.39 ± 8.21^ab^	88.20 ± 7.77^ab^	86.09 ± 7.29^b^	0.0004	0.0043	0.0386
TC (mg/dL)		197.32 ± 32.81^b^	206.68 ± 32.38^ab^	201.93 ± 31.89^ab^	209.23 ± 31.50^a^	0.0105	0.0068	0.0042
TG (mg/dL)		129.44 ± 80.70^b^	110.77 ± 54.57^ab^	111.88 ± 58.32^ab^	110.41 ± 51.93^b^	0.0390	0.0191	0.0740
HDL-C (mg/dL)		52.85 ± 12.14^a^	59.08 ± 14.43^a^	56.55 ± 13.58^ab^	58.65 ± 14.04^a^	0.0108	0.0009	0.0098
LDL-C (mg/dL)		118.45 ± 28.16^b^	128.56 ± 31.45^ab^	123.02 ± 31.69^ab^	128.92 ± 28.70^a^	0.0367	0.0166	0.0141
TC/HDL-C		3.89 ± 0.91	3.73 ± 0.99	3.76 ± 0.92	3.73 ± 0.93	0.2410	0.4858	0.8169
LDL-C/HDL-C		2.35 ± 0.75	2.32 ± 0.82	2.29 ± 0.77	2.33 ± 0.76	0.8302	0.9406	0.9696
TG/HDL-C		2.69 ± 1.80^a^	2.09 ± 1.39^b^	2.20 ± 1.52^ab^	2.06 ± 1.38^b^	0.0039	0.0033	0.0213

Values are presented as mean ± SD for continuous variables or as a numerical proportion for categorical variables. ^1^
*p* for trend was calculated by analysis of variance (ANOVA) (Jonckheere–Terpstra) and χ^2^ (Cochran–Armitage) tests. ^2^
*p* was calculated by ANOVA and χ^2^ tests. Means with the same letter are not significantly different in row by Bonferroni post hoc test (*p* > 0.05). ^3^ Adjusted for estradiol, age, and body weight. **^†^** Welch’s ANOVA. *Adj.*, adjusted *p* value; KI, Kupperman index; MRS, menopause rating scale; BMI, body mass index; WC, waist circumference; HC, hip circumference; WHR, waist to hip ratio; SBP, systolic blood pressure; DBP, diastolic blood pressure; TC, total cholesterol; TG, triglyceride; HDL-C, high-density lipoprotein-cholesterol; LDL-C, low-density lipoprotein-cholesterol.

**Table 2 jcm-09-01161-t002:** Comparison of laboratory characteristics by quartiles of follicle-stimulating hormone.

Variables	Follicle-Stimulating Hormone (mIU/mL)				*p* for Trend ^1^	*p* ^2^	*Adj. P* ^3^
	Q1 (≤62.20)	Q2 (62.21~76.87)	Q3 (76.88~94.00)	Q4 (≥94.01)			
WBC (4.8–10.8 × 10^3^/μL)	5.12 ± 1.22	4.83 ± 1.20	4.99 ± 1.16	4.87 ± 1.13	0.2567	0.1164	0.2533
RBC (4.2–5.4 × 100^3^/μL)	4.38 ± 0.29^a^	4.28 ± 0.31^b^	4.36 ± 0.32^ab^	4.33 ± 0.29^ab^	0.5826	0.0417	0.0629
Hemoglobin (12–16 g/dL)	13.42 ± 0.82^a^	13.16 ± 0.87^b^	13.33 ± 0.87^ab^	13.23 ± 0.76^ab^	0.1552	0.0373	0.0866
Hematocrit (37–47%)	39.84 ± 2.16	39.26 ± 2.58	39.84 ± 2.54	39.39 ± 2.06	0.3519	0.0664 ^†^	0.0804
Platelet (130–450 × 10^3^/μL)	249.57 ± 53.80	239.80 ± 42.60	238.46 ± 49.55	245.80 ± 55.65	0.6414	0.1864	0.3450
ALP (45–129 IU/L)	74.57 ± 18.10	74.71 ± 16.13	72.43 ± 15.68	76.39 ± 17.94	0.6073	0.2460	0.1460
γ-GTP (8–48 IU/L)	19.95 ± 14.17	18.20 ± 12.04	19.28 ± 13.67	16.84 ± 8.55	0.2659	0.1341	0.2910
AST (12–33 IU/L)	23.52 ± 5.79	23.12 ± 5.64	23.45 ± 5.39	22.59 ± 4.67	0.3413	0.4187	0.6417
ALT (5–35 IU/L)	22.68 ± 10.43	21.03 ± 8.26	21.80 ± 8.69	20.36 ± 6.65	0.1321	0.1042	0.3561
Total bilirubin (0.2–1.2 mg/dL)	0.81 ± 0.30	0.84 ± 0.27	0.81 ± 0.23	0.82 ± 0.26	0.6695	0.7025	0.7281
Total protein (6.7–8.3 g/dL)	7.35 ± 0.34	7.28 ± 0.31	7.29 ± 0.35	7.29 ± 0.38	0.2343	0.6317	0.6231
Albumin (3.5–5.3 g/dL)	4.38 ± 0.19	4.37 ± 0.18	4.39 ± 0.18	4.38 ± 0.20	0.7952	0.3864	0.4043
BUN (8–23 mg/dL)	14.61 ± 3.93	14.54 ± 3.25	14.98 ± 3.51	14.55 ± 3.28	0.4111	0.6553	0.6190
eGFR (90–120 mL/min/1.73 m^2^)	114.15 ± 19.67	112.44 ± 23.43	111.44 ± 22.26	110.08 ± 18.76	0.1079	0.3909	0.0970
Uric acid (2.5–6.3 mg/dL)	4.56 ± 1.13	4.36 ± 0.92	4.56 ± 0.80	4.37 ± 0.99	0.8558	0.5556	0.4102
Total calcium (8.4–10.2 mg/dL)	9.46 ± 0.33	9.55 ± 0.36	9.57 ± 0.34	9.52 ± 0.32	0.2884	0.1458	0.0689
TSH (0.55–4.78 μIU/mL)	1.80 ± 0.93	1.81 ± 1.01	1.94 ± 0.85	1.97 ± 1.02	0.0514	0.4016	0.3014
hs-CRP (~5 mg/L)	0.92 ± 3.63	0.50 ± 1.31	0.41 ± 0.85	1.00 ± 2.46	0.2563	0.4845	0.5692
Urine specific gravity (1.005–1.030)	1.02 ± 0.01	1.02 ± 0.01	1.02 ± 0.00	1.02 ± 0.01	0.7690	0.1507	0.1298
Urine pH (4.5–9.0)	6.05 ± 0.85	6.16 ± 0.86	6.18 ± 0.85	6.14 ± 0.86	0.2979	0.5405	0.6468

Values are presented as mean ± SD for continuous variables or as a numerical proportion for categorical variables. ^1^
*p* for trend was calculated by ANOVA (Jonckheere-Terpstra) tests. ^2^
*p* was calculated by ANOVA. Means with the same letter are not significantly different in row by Bonferroni post hoc test (*p* > 0.05). ^3^ Adjusted for estradiol, age, and body weight. ^†^ Welch’s ANOVA. *Adj.*, adjusted *p* value; ALP, alkaline phosphatase; γ-GTP, γ-glutamyl transferase; AST, aspartate transferase; ALT, alanine transferase; BUN, blood urea nitrogen; eGFR, estimated glomerular filtration rate; TSH, thyroid stimulating hormone; hs-CRP, high sensitivity C-reactive protein.

**Table 3 jcm-09-01161-t003:** Comparisons of metabolic syndrome and number of its components by quartiles of follicle-stimulating hormone.

Variables	Follicle-Stimulating Hormone (mIU/mL)				*p* for Trend ^1^	*P* ^2^	*Adj. P* ^3^
	Q1 (≤62.20)	Q2 (62.21~76.87)	Q3 (76.88~94.00)	Q4 (≥94.01)			
MetS (N, %)	29(33.33%)	17(18.68%)	19(24.68%)	11(15.71%)	0.0281	0.0402	-
Number of the component of MetS	2.01 ± 1.06^a^	1.45 ± 1.20^b^	1.51 ± 1.15^b^	1.41 ± 1.07^b^	0.0015	0.0015	0.0215

Values are presented as mean ± SD for continuous variables or as a numerical proportion for categorical variables. ^1^
*p* for trend was calculated by ANOVA (Jonckheere–Terpstra) and χ^2^ (Cochran–Armitage) tests. ^2^
*p* was calculated by ANOVA and χ^2^ tests. Means with the same letter are not significantly different in row by Bonferroni post hoc test (*p* > 0.05). ^3^ Adjusted for estradiol, age, and body weight. *Adj.*, Adjusted *p* value; MetS; metabolic syndrome.

**Table 4 jcm-09-01161-t004:** Associations between follicle-stimulating hormone and metabolic syndrome in post-menopausal women.

Variables	FSH	Unstandardized Coefficients (B)	Odds Ratio	*p*-Value	95% CI	
					Lower	Upper
**Metabolic syndrome**	Q1	0.872	2.391	0.039	1.047	5.459
	Q2	0.228	1.256	0.606	0.528	2.991
	Q3	0.710	2.035	0.109	0.853	4.853
	Q4		1.000			

Data were analyzed using logistic regression models with metabolic syndrome (Y/N) as the outcome and follicle-stimulating hormone (FSH) as the explanatory variable. The results were expressed as odds ratios (95% confidence interval). The model 2 was adjusted for age and estradiol. The model was adjusted for age, estradiol, and obesity.

**Table 5 jcm-09-01161-t005:** Associations between follicle-stimulating hormone and components of metabolic syndrome in post-menopausal women.

Variables	FSH	Unstandardized Coefficients (B)	Odds Ratio	*p*-Value	95% CI	
					Lower	Upper
Hypertension	Q1	0.436	1.546	0.123	0.888	2.690
	Q2	0.075	1.078	0.802	0.601	1.931
	Q3	0.057	1.058	0.849	0.591	1.896
	Q4		1.000			
WC (central obesity)	Q1	0.958	2.606	0.024	1.136	5.975
	Q2	0.215	1.240	0.571	0.589	2.609
	Q3	0.175	1.191	0.650	0.560	2.535
	Q4		1.000			
HDL-C	Q1	0.466	1.593	0.095	0.921	2.754
	Q2	−0.185	0.832	0.537	0.463	1.494
	Q3	0.160	1.173	0.578	0.668	2.060
	Q4		1.000			

Data were analyzed using logistic regression models with components of metabolic syndrome (Y/N) as the outcome and follicle-stimulating hormone (FSH) as the explanatory variable. The results were expressed as odds ratios (95% confidence interval). The model was adjusted for age, estradiol, and obesity.

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
