# Peer review of "Serum Follicle-Stimulating Hormone Levels are Associated with Cardiometabolic Risk Factors in Post-Menopausal Korean Women"

_jcm, 2020, doi:10.3390/jcm9041161_

Round 1

Reviewer 1 Report

Jung et al. investigated the association between serum FSH levels and cardiometabolic risk factors in postmenopausal Korean women.

The authors found that the risk for metabolic syndrome (MetS) decreased as FSH quartiles increased. FSH levels were negatively associated with cardiometabolic risk factors including BMI, body weight, waist circumference, fasting glucose and TG, while it was positively associated with HDL-C. The authors concluded that FSH levels had a negative correlation with cardiometabolic risk factors in postmenopausal Korean women, suggesting that a low FSH can be a predictor for CVD in this population.

Pros: The study comprised of a fairly large pool of patients.

Comments:

Are there data on HbA1c available? This could be of special interest.

The authors state that patients having any other current illnesses or any history of chronic medical diseases within the past 3 years were not included. How was this determined? Testing for HbA1c would be of interest to rule out diabetes. How were patients with hypertension included? Did patients take medication on a regular basis, if yes which kind of medication (anti-hypertensive drugs, statins)? I would recommend to define the patient population better, stating possible previous diseases and/or medication intake.

Reviewer 2 Report

In the article entitled: Serum follicle-stimulating hormone levels are associated with cardiometabolic risk factors in postmenopausal Korean women the Authors assessed association between follicle-stimulating hormone (FSH) and cardiometabolic risks in postmenopausal women. This is an original and interesting paper providing the reader to the new issue of atherosclerosis pathophysiology. Women’s cardiometabolic risk significantly increases after shifting into menopause but associations between endogenous estrogens and metabolic cardiovascular risk were not consistent. Even less is known about the role of FSH in the development and progression of atherosclerotic cardiovascular disease.

The manuscript is well written. Nevertheless, to become suitable for publication, the Authors should raise the manuscript profile adding some relevant additional details and comments.

  1. Chapter 2.2. Biochemical and hormone analyses does not mention TSH, which was later rightly included in the statistical analysis.
  2. The waist circumference was measured. Why was not given and included in the analysis waist to hip ratio. This is an extremely important factor in the assessment of metabolic disorders, especially in the metabolic syndrome also analyzed in the study.
  3. In assessing renal function, the use of eGFR rather than creatinine levels is more objective..
  4. How about thyroid dysfunction (hyperthyroidism hypothyroidism)? This is common, especially hypothyroidism in postmenopausal women. The “Flow diagram of study selection process” (figure 1) does not state that the abnormal thyroid function was the exclusion criterion. Thus, it can be considered that this was not taken into account. This would be inappropriate in a study dedicated to cardiometabolic risk factors in postmenopausal women.  
  5. Enter the range of normal values in the laboratory for at least FSH and estradiol.
  6. In Table 1 “General characteristics of study participants” specify the average values of the whole study group for all parameters which are then analyzed divided into quartiles in Table 2 and 3.
  7. How the division into quartiles was made - a detailed description is necessary, especially since it is the basis for further analysis.
  8. In line 148 authors wrote: General demographic characteristics were categorized according to FSH quartiles (Table 2). A table shows the level of estradiol. Is the estradiol a demographic indicator? I don't think so.
  9. Why are the other cardiometabolic risk factors: total cholesterol, LDL-cholesterol not included in the linear regression analysis in Figure 2?
  10. It would be advised to carry out additional, similar to that shown in Figure 2 the analysis of estradiol instead of FSH: next figure - “Associations between estradiol and cardiometabolic risk factors adjusted for age, follicle-stimulating hormone, and body weight in postmenopausal women”. After all, estradiol is an effector hormone, and basically it, and not FSH, directly affects metabolism. There is an increased prevalence of dyslipidemia and cholesterol accumulation around the menopausal transition, which has traditionally been linked to estrogen deficiency. However, there has been recent evidence that FSH might play a role in increasing hepatic cholesterol production, independent of serum estrogen.
  11. Both “associations between follicle-stimulating hormone and cardiometabolic risk factors” (figure 2) and any additional analysis regarding “associations between estradiol and cardiometabolic risk factors” should be adjusted to TSH. TSH as a marker of thyroid function can significantly change the results of both analyzes in the context of the effect of FSH and estradiol on metabolic factors.
  12. In discussion (line 238-241) authors have stated that: In this study, FSH levels, but not estrogen, of the postmenopausal women had a significant relation to cardiometabolic risk factors and the prevalence of MetS or its components (Table 7, 8 and Figure 2). I don't think that's right, because figure 2 concerns:  “Associations between follicle-stimulating hormone and cardiometabolic risk factors” only with estradiol adjustment. If the same results can be achieved for estradiol it would be to agree with the position of authors. I suggest adding the next figure mentioned in point 10 of the review - “Associations between estradiol and cardiometabolic risk factors adjusted for age, follicle-stimulating hormone, and body weight in postmenopausal women”.
  13. Literature is not sufficiently up to date. There have been many recent papers published, e.g.:

Guo, Y.; Zhao, M.; Bo, T.; Ma, S.; Yuan, Z.; Chen, W.; He, Z.; Hou, X.; Liu, J.; Hang, Z.; et al. Blocking FSH inhibits hepatic cholesterol biosynthesis and reduces serum cholesterol. Cell Research. 2019;29:151–166; https://doi.org/10.1038/s41422-018-0123-6.

Author Response

In the article entitled: “Serum follicle-stimulating hormone levels are associated with cardiometabolic risk factors in postmenopausal Korean women” the Authors assessed association between follicle-stimulating hormone (FSH) and cardiometabolic risks in postmenopausal women. This is an original and interesting paper providing the reader to the new issue of atherosclerosis pathophysiology. Women’s cardiometabolic risk significantly increases after shifting into menopause but associations between endogenous estrogens and metabolic cardiovascular risk were not consistent. Even less is known about the role of FSH in the development and progression of atherosclerotic cardiovascular disease.

The manuscript is well written. Nevertheless, to become suitable for publication, the Authors should raise the manuscript profile adding some relevant additional details and comments.

1. Chapter 2.2. Biochemical and hormone analyses does not mention TSH, which was later rightly included in the statistical analysis.

--> We described analytical method for TSH as follows:

Serum TSH was measured by ADVIA Centaur XP Immunoassay System (Siemens Healthcare GmbH, Germany).

2. The waist circumference was measured. Why was not given and included in the analysis waist to hip ratio. This is an extremely important factor in the assessment of metabolic disorders, especially in the metabolic syndrome also analyzed in the study.

--> As suggested, we included waist to hip ratio in Table S1. We described this point in the Results section as follows:

Overall, the study participants were characterized by postmenopausal women having moderate menopausal symptoms with overweight (23.54±2.69 kg/m2 of BMI), central obesity (83.32±7.72 cm of WC 0.88±0.06 of waist to hip ratio), and dyslipidemia (203.78±32.40 mg/dl of total cholesterol levels).

3. In assessing renal function, the use of eGFR rather than creatinine levels is more objective.

--> As suggested, in Table 2, we used eGFR instead of creatinine to assess renal function.

4. How about thyroid dysfunction (hyperthyroidism hypothyroidism)? This is common, especially hypothyroidism in postmenopausal women. The “Flow diagram of study selection process” (figure 1) does not state that the abnormal thyroid function was the exclusion criterion. Thus, it can be considered that this was not taken into account. This would be inappropriate in a study dedicated to cardiometabolic risk factors in postmenopausal women.

--> We analyzed the association between FSH and cardiometabolic risk factors after adjustment for TSH and added data in Figure 2B. We further analyzed the association between estrogen and cardiometabolic risk factors after adjustment for TSH and added data in Figure 3B (see Response to comment 11).

5. Enter the range of normal values in the laboratory for at least FSH and estradiol.

--> We included normal values of FSH and estradiol in the Results section as follows:

The mean serum FSH level was 79.67±24.36 mIU/ml (reference ranges: 25.8-134.8 mIU/ml) and estradiol level (reference ranges: 5.0-138.0 pg/ml) was 7.99±9.38 pg/ml.

6. In Table 1 “General characteristics of study participants” specify the average values of the whole study group for all parameters which are then analyzed divided into quartiles in Table 2 and 3.

--> As commented, we included average values of the anthropometric and laboratory parameters in Tables S1.

7. How the division into quartiles was made - a detailed description is necessary, especially since it is the basis for further analysis.Participants were divided into quartiles according to the serum FSH levels, with an equal number of participants in each group. A Chi-square (χ2) test was then performed to determine differences between the frequencies of categorized variables between the FSH quartiles groups.

--> The quartiles of a data set divide the data into four equal parts, with one-fourth of the data values in each part. We described this point in the Methods section as follows:

Participants were divided into quartiles according to the serum FSH levels, with an equal number of participants in each group.

8. In line 148 authors wrote: General demographic characteristics were categorized according to FSH quartiles (Table 2). A table shows the level of estradiol. Is the estradiol a demographic indicator? I don't think so.General demographic anthropometric and laboratory characteristics were categorized according to FSH quartiles (Table 1).

--> We thank you for your correction.

General demographic anthropometric and laboratory characteristics were categorized according to FSH quartiles (Table 1).

9. Why are the other cardiometabolic risk factors: total cholesterol, LDL-cholesterol not included in the linear regression analysis in Figure 2?

--> As commented, we included total cholesterol and LDL-cholesterol in the linear regression analysis.

10. It would be advised to carry out additional, similar to that shown in Figure 2 the analysis of estradiol instead of FSH: next figure - “Associations between estradiol and cardiometabolic risk factors adjusted for age, follicle-stimulating hormone, and body weight in postmenopausal women”(Figure 3). After all, estradiol is an effector hormone, and basically it, and not FSH, directly affects metabolism. There is an increased prevalence of dyslipidemia and cholesterol accumulation around the menopausal transition, which has traditionally been linked to estrogen deficiency. However, there has been recent evidence that FSH might play a role in increasing hepatic cholesterol production, independent of serum estrogen.

--> As commented, we further analyzed “associations between estradiol and cardiometabolic risk factors after adjustment for age, FSH, and body weight in postmenopausal women”. We described this point in the Results section as follows:

Association between estradiol levels and cardiometabolic risk factors

Because estrogen is the primary hormone responsible for CVD in menopausal women [9], we further analyzed the association between estrogen levels and cardiometabolic risk factors. Estradiol level was inversely related with WC (B -0.08; 95% CI, -0.038 to -0.119, P<0.05) after adjustment for age, body weight, and FSH (Figure 3A). When the analyses were repeated after adjustment for age, body weight, and TSH, estradiol level was positively correlated with body weight (B 0.255; 95% CI, 0.111 to 0.399, P<0.001) and BMI (B 0.096; 95% CI, 0.044 to 0.148, P<0.001) and inversely related with HDL-C (B -0.437; 95% CI, -0.793 to -0.080, P<0.05) (Figure 3B).

11. Both “associations between follicle-stimulating hormone and cardiometabolic risk factors” (figure 2) and any additional analysis regarding “associations between estradiol and cardiometabolic risk factors” should be adjusted to TSH. TSH as a marker of thyroid function can significantly change the results of both analyzes in the context of the effect of FSH and estradiol on metabolic factors.

--> As commented, we further analyzed “associations between FSH and cardiometabolic risk factors after adjustment for age, estradiol, and body weight in postmenopausal women” in Figure 2B. We also analyzed “associations between estradiol and cardiometabolic risk factors after adjustment for age, TSH, and body weight in postmenopausal women” in Figure 3B. We described this point in the Results section as follows:

Elevation of serum thyroid stimulating hormone (TSH) is common in postmenopausal women [24], and serum levels of TSH associate positively with cardiometabolic risk factors even in postmenopausal women with normal thyroid function [25, 26]. We therefore analyzed the association between FSH and cardiometabolic risk factors after adjustment for age, body weight, and TSH. Results show that FSH was also inversely related with cardiometabolic risk factors: BMI (B -0.028; 95% CI, -0.038 to -0.017, P<0.001), body weight (B -0.077; 95% CI, -0.105 to -0.048, P<0.001), WC (B -0.035; 95% CI, -0.059 to -0.011, P<0.01), and fasting glucose (B -0.031; 95% CI, -0.062 to 0.001, P=0.055), while the FSH level was positively correlated with total cholesterol (B 0.178; 95% CI, 0.042 to 0.313, P<0.05) and LDL-C (B 0.164; 95% CI, 0.013 to 0.315, P<0.05) (Figure 2B). Overall, the results indicated that higher FSH level was associated with lower cardiometabolic risk factors. 

12. In discussion (line 238-241) authors have stated that: In this study, FSH levels, but not estrogen, of the postmenopausal women had a significant relation to cardiometabolic risk factors and the prevalence of MetS or its components (Table 7, 8 and Figure 2). I don't think that's right, because figure 2 concerns:  “Associations between follicle-stimulating hormone and cardiometabolic risk factors” only with estradiol adjustment. If the same results can be achieved for estradiol it would be to agree with the position of authors. I suggest adding the next figure mentioned in point 10 of the review - “Associations between estradiol and cardiometabolic risk factors adjusted for age, follicle-stimulating hormone, and body weight in postmenopausal women”.

--> We thank you for your correction and suggestion. We rephrased the above sentence as follows:

In this study, FSH, but not estrogen, and estrogen levels of the postmenopausal women had a significant relation to cardiometabolic risk factors and the prevalence of MetS or its components (Tables 4, 5 and Figure 2).

13. Literature is not sufficiently up to date. There have been many recent papers published, e.g.:Guo, Y.; Zhao, M.; Bo, T.; Ma, S.; Yuan, Z.; Chen, W.; He, Z.; Hou, X.; Liu, J.; Hang, Z.; et al. Blocking FSH inhibits hepatic cholesterol biosynthesis and reduces serum cholesterol. Cell Research. 2019;29:151–166; https://doi.org/10.1038/s41422-018-0123-6.

--> We cited the above article in the Discussion section as follows:

In addition, FSH is known to increase neovascularization in the umbilical vein [39] and inhibits hepatic cholesterol biosynthesis [40] via the direct effect, and indirectly in granulosa cells [41] and ovarian cancer cells [42] by inducing vascular endothelial growth factor (VEGF) production.

Round 2

Reviewer 1 Report

The authors responed to all questions.

Reviewer 2 Report

Accept in present form